# Peroxisome Proliferator-Activated Receptor γ Coactivator 1α Activates Vascular Endothelial Growth Factor That Protects Against Neuronal Cell Death Following Status Epilepticus through PI3K/AKT and MEK/ERK Signaling

**DOI:** 10.3390/ijms21197247

**Published:** 2020-09-30

**Authors:** Jyun-Bin Huang, Shih-Pin Hsu, Hsiu-Yung Pan, Shang-Der Chen, Shu-Fang Chen, Tsu-Kung Lin, Xuan-Ping Liu, Jie-Hau Li, Nai-Ching Chen, Chia-Wei Liou, Chung-Yao Hsu, Hung-Yi Chuang, Yao-Chung Chuang

**Affiliations:** 1Department of Emergency Medicine, Kaohsiung Chang Gung Memorial Hospital, Kaohsiung 83301, Taiwan; u9001135@gmail.com (J.-B.H.); fornever@cgmh.org.tw (H.-Y.P.); 2College of Medicine, Chang Gung University, Taoyuan 33302, Taiwan; chensd@adm.cgmh.org.tw (S.-D.C.); fangoe1@yahoo.com.tw (S.-F.C.); tklin@adm.cgmh.org.tw (T.-K.L.); naiging@yahoo.com.tw (N.-C.C.); cwliou@ms22.hinet.net (C.-W.L.); 3Department of Neurology, E-Da Hospital/School of Medicine, I-Shou University, Kaohsiung 824, Taiwan; a.pin.hsu@gmail.com; 4Department of Neurology, Kaohsiung Chang Gung Memorial Hospital, Kaohsiung 83301, Taiwan; 5Institute for Translation Research in Biomedicine, Kaohsiung Chang Gung Memorial Hospital, Kaohsiung 83301, Taiwan; pphome67@yahoo.com.tw (X.-P.L.); jiehau1060301@gmail.com (J.-H.L.); 6Mitochondrial Research Unit, Kaohsiung Chang Gung Memorial Hospital and Chang Gung University College of Medicine, Kaohsiung 83301, Taiwan; 7Department of Neurology, School of Medicine, College of Medicine, Kaohsiung Medical University Hospital, Kaohsiung Medical University, Kaohsiung 80708, Taiwan; cyhsu61@gmail.com; 8Department of Occupational and Environmental Medicine, Kaohsiung Medical University Hospital and School of Public Health, Kaohsiung Medical University, Kaohsiung 80708, Taiwan; hychuang@gmail.com; 9Department of Biological Science, National Sun Yat-sen University, Kaohsiung 80424, Taiwan

**Keywords:** neuroprotection, PGC-1α, vascular endothelial growth factor, vascular endothelial growth factor receptor 2, PI3K/AKT, MEK/ERK, status epilepticus, hippocampus

## Abstract

Status epilepticus may cause molecular and cellular events, leading to hippocampal neuronal cell death. Peroxisome proliferator-activated receptor γ coactivator 1-α (PGC-1α) is an important regulator of vascular endothelial growth factor (VEGF) and VEGF receptor 2 (VEGFR2), also known as fetal liver kinase receptor 1 (Flk-1). Resveratrol is an activator of PGC-1α. It has been suggested to provide neuroprotective effects in epilepsy, stroke, and neurodegenerative diseases. In the present study, we used microinjection of kainic acid into the left hippocampal CA3 region in Sprague Dawley rats to induce bilateral prolonged seizure activity. Upregulating the PGC-1α pathway will increase VEGF/VEGFR2 (Flk-1) signaling and further activate some survival signaling that includes the mitogen activated protein kinase kinase (MEK)/mitogen activated protein kinase (ERK) and phosphatidylinositol 3-kinase (PI3K)/protein kinase B (AKT) signaling pathways and offer neuroprotection as a consequence of apoptosis in the hippocampal neurons following status epilepticus. Otherwise, downregulation of PGC-1α by siRNA against *pgc-1α* will inhibit VEGF/VEGFR2 (Flk-1) signaling and suppress pro-survival PI3K/AKT and MEK/ERK pathways that are also accompanied by hippocampal CA3 neuronal cell apoptosis. These results may indicate that the PGC-1α induced VEGF/VEGFR2 pathway may trigger the neuronal survival signaling, and the PI3K/AKT and MEK/ERK signaling pathways. Thus, the axis of PGC-1α/VEGF/VEGFR2 (Flk-1) and the triggering of downstream PI3K/AKT and MEK/ERK signaling could be considered an endogenous neuroprotective effect against apoptosis in the hippocampus following status epilepticus.

## 1. Introduction

Epilepsy is one of the most common serious brain conditions characterized by the recurrence of unprovoked seizures, affecting more than 70 million people worldwide [1]. Status epilepticus is a common neurological and medical emergency, which is associated with high morbidity, mortality, and health-care burden [1]. Sustained seizure activities during status epilepticus usually result in significant neuronal damage in the cerebral cortex, particularly in the hippocampus [2,3]. Evidence has shown both in human and animal studies that prolonged seizures may lead to a large number of changes of molecular and cellular cascades, including axonal sprouting, gliosis, network reorganization, activation of neuroinflammation, acquired channelopathies, oxidative stress, mitochondrial dysfunctions, angiogenesis, neurogenesis, and activation of some late cell death pathways, which contribute to neurodegeneration and brain damage [4,5,6,7]. 

Vascular endothelial growth factors (VEGFs) and their receptors (VEGFRs) have important roles in the formation, function, and maintenance of blood vessels and are essential regulators of angiogenesis and vascular permeability [8]. Recent evidence have shown that VEGFs also have crucial roles in other organ systems, including the central nervous system (CNS), kidney, lung, and liver, where they directly influence organ function and development [9]. Among the VEGF family, VEGF-A is currently considered to play an important role in neuroprotective effects on hypoxic motor neurons and amyotrophic lateral sclerosis [8]. The physiological functions of VEGF are majorly mediated by the receptors of VEGF (VEGFRs). VEGF-A can bind to and activate two tyrosine kinase receptors, including VEGFR-1 and VEGFR-2 [8]. VEGFR-2 is also referred to as the fetal liver kinase receptor 1 (Flk-1) [9], which mediates most of the endothelial growth and survival signals [8]. In addition, VGEFR2 has been reported on glia and neurons where they might be upregulated during neuronal perturbations [10,11,12,13]. In the nervous system, the most important VEGF receptor pathway is VEGF receptor 2 (VEGFR2) [9,10,11]. Under perturbations of neuronal cells, availability of VEGFA to bind to VGEFR2 (Flk-1) will increase and promote VGEFR2 (Flk-1) downstream signaling [9,12,14]. Recent evidence suggests that VEGF has therapeutic potential as a neuroprotective factor in many neurological diseases [12,15], such as stroke [16,17,18], epilepsy [14,19], and neurodegenerative diseases [12,20,21,22].

Both in the animal studies following sustained seizure activities [14,19,23], and in resected tissues from patients with focal cortical dysplasia and refractory epilepsy [24], VEGF is up-regulated in neural cells and glia in the hippocampus and pyramidal neurons of the cortex. In animal models, overexpression of VEGF mRNA and protein and blood-brain barrier (BBB) impairment in the hippocampus occurred early after electroconvulsive shock-induced seizures or pilocarpine-induced seizures [23,24,25]. The neuroprotective role of VEGF in ischemic stroke has been well studied. In epileptic seizures, such as status epilepticus, the literature on the neuroprotective role of the VEGF-A/VEGFR2 pathway is limited. Peroxisome proliferator-activated receptor γ (PPARγ) coactivator 1-α (PGC1-α) belongs to a small family of transcriptional coactivators identified as a cofactor for the nuclear hormone receptor PPARγ that possesses a common function in mitochondrial physiology and mitochondrial biogenesis [5,26]. Additionally, it is also involved in other metabolic processes, such as redox homeostasis, uncoupled respiration, and gluconeogenesis [26,27,28]. Hypoxia-induced upregulation of VEGF mRNA is associated with increases in the transcription factor hypoxia inducible factor (HIF)1-α, which is also upregulated after cerebral ischemia [11,18].

The trigger for increased neuronal VEGF after seizures and brain insults is unclear. Beyond well-studied VEGF regulator HIF-1α [11,18], activation of PGC-1α induces the expression of VEGF signaling, leading to the formation of new blood vessels and protecting from cell damage, in an HIF-1α independent pathway [29,30,31]. Therefore, PGC-1α in the regulation of VEGF/VEGFR2 signaling pathway in the neuronal cells might be a crucial mechanism in neuroprotection following status epilepticus.

Recent research showed that the upregulation of VEGF/VEGFR2 (Flk-1) occurred after sustained seizures and VEGF signaling offered neuroprotective effects against neuronal cell death in the hippocampus following status epilepticus [14,19,23]. When upregulation of VEGF in neurons and glial cells occurs after persistent epileptic seizures, it counteracts seizure-induced neurodegeneration via overexpression of VEGFR2 (Flk-1) [14]. A previous study [14] suggested that increased VEGF signaling pathway via overexpression of VEGFR2 may affect seizure activity even without altering angiogenesis. Therefore, VEGFR2 could be considered a novel target for developing therapy strategies against epileptic activities [14]. However, whether the VEGF/VEGFR2 (Flk-1) signaling pathway contributes neurogenesis and is counteractive to epileptogenesis remains unclear [14]. Growing evidence suggests that under stressful stimuli, such as prolonged epileptic seizures, the VEGF/VEGFR2 (Flk-1) signaling pathway may participate in mediating angiogenesis, neuronal migration, hippocampal cell proliferation, and anti-apoptosis, which provide neuroprotective effects through the downstream activation of survival signaling, including phosphatidylinositol 3-kinase (PI3K)/protein kinase B (AKT) and the mitogen activated protein kinase kinase (MEK)/mitogen activated protein kinase (ERK) [13,14,32,33].

In the present study, we propose that PGC-1α may be activated during experimental status epilepticus and regulate the VEGF/VEGFR2 signaling pathway, and further protects against apoptotic neuronal cell death in the hippocampus following status epilepticus through survival signaling and PI3K/AKT and MEK/ERK-dependent pathways.

## 2. Results

### 2.1. Temporal Changes of PGC-1α Expression in the Hippocampal CA3 following Status Epilepticus

Temporal changes of PGC-1α expression in the right hippocampal CA3 region were examined following status epilepticus. After unilateral microinjection of kainic acid (KA; 0.5 nmol) into the left CA3 subfield, *pgc-1α* mRNA expression exhibited a significantly incremental change in the right hippocampal CA3 area 1 h after the elicitation of prolonged seizure activities, followed by a progressive decrement that returned to base line at 24 h (Figure 1a). For detection of the PGC-1α protein level, western blot analysis also presented a significant augment of PGC-1α protein in the right hippocampal CA3 subfield 1–24 h after the status epilepticus induced by KA (Figure 1b).

### 2.2. Temporal Changes of VEGF and VEGFR2 Expression in the Hippocampal CA3 Region following Status Epilepticus

The expression of VEGF and VEGFR2 following KA-induced experimental status epilepticus were examined. Real-time PCR results showed the expression of *vegf* mRNA extracted from the right hippocampal CA3 region was significantly augmented at 3 h, 6 h, and 24 h after administration of KA (Figure 2a). Correspondingly, western blotting analysis also showed a significant increase of VEGF protein levels at 3, 6, and 24 h after experimental status epilepticus that peaked at 6 h (Figure 2b). Additionally, in the parallel experiments, the expression of *vegfr2* mRNA also exhibited a significant increase from 3 to 24 h after KA-induced status epilepticus (Figure 2c). By western blotting analysis, the VEGFR2 protein levels were revealed to be significantly increased in the right hippocampal CA3 tissue 3–24 h after the induction of status epilepticus (Figure 2d).

### 2.3. Effect of Resveratrol and Gene Knock-Down by Small Interfering RNA (siRNA) Against pgc-1α on VEGF Expression in the Hippocampus Following Experimental Status Epilepticus

In a previous report [26], we demonstrated that resveratrol is a PGC-1α activator, and it can activate PGC-1α and the following signaling pathways and then promote mitochondrial biogenesis. To determine the causality of PGC-1α in modulation of VEGF on this experimental paradigm, we tested the effects of resveratrol, as a positive control, and siRNA against *pgc-1α,* as a negative control, on VEGF expression in the hippocampus following status epilepticus. Microinjection of resveratrol (100 μmol) into the hippocampal CA3 region significantly increased the expression of *vegf* mRNA (Figure 3a) and the level of VEGF protein (Figure 3b) in the CA3 region 6 h after the elicitation of sustained hippocampal seizure activities. On the other hand, the specificity of the gene knock-down strategy by siRNA against *pgc-1α* was tested for VEGF expression in the hippocampus. For confirming the specific effect of *pgc-1α* siRNA on VEGF expression, after a pre-treated microinjection with siRNA against *pgc-1α* (2 μg) into bilateral hippocampal CA3 region, significant decrease of the *vegf* mRNA level was shown in our real-time PCR data (Figure 3c), and western blot analysis also confirmed a drastically decline in the VEGF protein level in the hippocampal CA3 area 6 h after KA-induced status epilepticus (Figure 3d) compared with the pre-treatment of the control siRNA group.

To confirm that the changes in expression of VEGF by resveratrol treatment and *pgc-1α* gene knock-down observed in our above biochemical analyses, we applied double immunofluorescence staining to detect the intracellular expression of VEGF in hippocampal CA3 neurons (Figure 4). Within the unified fields of a laser scanning confocal microscope, signals for VEGF were weak in the neurons, which were strongly immunoreactive to the neuron-specific nuclear protein (NeuN) neuronal marker, of hippocampal CA3 in the control animals (Figure 4a–c). In contrast to the control, there was an increase in VEGF immunoreactivity in the neurons from the same field, the CA3 area, 6 h after KA-induced status epilepticus (Figure 4d–f). Pretreatment of resveratrol (100 μmol) augmented the PGC-1α immunoreactivity in the hippocampal CA3 neurons (Figure 4j–l).

In addition, in the parallel experiments, many VEGF-positive neurons were detected in the hippocampal CA3 area 6 h after experimental status epilepticus in animals pre-treated with pgc-1α control siRNA. However, reduced VEGF-positive cells were observed in the hippocampal CA3 subfield 6 h after status epilepticus in rats with pre-treated microinjection of control siRNA (Figure 4j–i) or siRNA against pgc-1α (2 μg) (Figure 4m–o) into the bilateral hippocampal CA3 field.

### 2.4. Effect of Resveratrol and Gene Knock-Down by Small Interfering RNA (siRNA) Against pgc-1α on Expression of VEGF Receptor 2 (VEGFR2) in the Hippocampus Following Experimental Status Epilepticus

To determine the causal effect of PGC-1α in regulation of the expression of VEGFR2 in this experimental paradigm, we tested resveratrol, a PGC-1α activator, and siRNA against *pgc-1α* on VEGF expression on VEGFR2 expression in the hippocampus following status epilepticus. Microinjection of resveratrol (100 μmol) into the hippocampal CA3 subfield underwent a significant increment of expression of *vegf* mRNA (Figure 5a) and the level of VEGFR2 protein (Figure 5b) in the CA3 region 6 h after the elicitation of sustained hippocampal seizure activities. Confirming the specificity of *pgc-1α* siRNA on the protein levels of VEGFR2, after a pre-treated microinjection of *pgc-1α* siRNA (2 μg) into the bilateral CA3 region, a significant elevation of the expression of *vegfr2* mRNA (Figure 5c) and diminishment of VEGF protein in the CA3 subfield 6 h after KA-induced status epilepticus (Figure 5d) compared with the rats with pre-treatment of control siRNA was observed.

### 2.5. Activation of VEGF/VEGFR2 Regulates the PI3K/Akt Survival Signaling Pathway in the Hippocampal Neurons Following Experimental Status Epilepticus

The activation of VEGF/VEGFR2 can subsequently activate PI3K/AKT signaling by altering the activation state of numerous downstream proteins relevant to cell proliferation and inhibiting apoptosis [34,35]. To determinate whether the PGC-1α regulates the survival signaling pathway, PI3K/Akt, through VEGF/VEGFR2 activation in the hippocampus following KA-induced status epilepticus, we delineated the regulatory effects of the PGC-1α activator and siRNA against *pgc-1α* on PI3K/Akt signaling. In the experiment, the phosphorylation state of PI3K (p-PI3K) and AKT (p-AKT), and total levels of PIK3 and AKT were measured by western blotting analysis. The ratio of p-PI3K/total PI3K and p-AKT/total AKT were raised 6 h after microinjection of 3% dimethyl sulfoxide (DMSO) and KA into the left hippocampal CA3 region compared with controls (Figure 6a,b).

In addition, the ratio of p-PI3K/total PI3K and p-AKT/total AKT were significantly enhanced 6 h after microinjection of resveratrol (100 μmol) followed by KA administration into the left hippocampal CA3 region (Figure 6a,b). On the other hand, the ratio of p-PI3K/total PI3K and p-AKT/total AKT were raised 6 h after administration of control siRNA with KA into left hippocampal CA3 region compared with the control group (Figure 6c,d). However, the ratio of p-PI3K/total PI3K and p-AKT/total AKT were inhibited in the hippocampal CA3 neuronal cells 24 h before pre-treated microinjection of siRNA against *pgc-1α* (2 μg) followed with KA microinjection into the hippocampus. Akt can be activated by p-PI3K and active PI3K (Figure 6c,d).

### 2.6. Activation VEGF/VEGFR2 Regulates the MEK/ERK Signaling Pathway in the Hippocampal Neurons Following Experimental Status Epilepticus

VEGF/VEGFR2 activates the MEK/ERK signaling pathway as evidenced by an increase in ERK and MEK phosphorylation levels [36,37]. To determine the regulatory role of VEGE/VEGFR2 in the MEK/ERK pathway, we examined the protein levels of phosphorylated ERK (p-ERK) and MEK (p-MEK), and total ERK and MEK. Rats received pre-treatment with resveratrol or siRNA against *pgc-1α* and these proteins were analyzed by western blotting 6 h after KA-induced experimental status epilepticus. The ratio of p-ERK/total ERK and p-MEK/total MEK were increased 6h after microinjection of DMSO and KA into the left hippocampal CA3 region compared with control animals (Figure 7a,b). Additionally, the ratios of p-ERK/total ERK and p-MEK/total MEK were significantly increased 6 h after microinjection of resveratrol (100 µmol) followed by KA administration into the left hippocampal CA3 (Figure 7a,b). Otherwise, the ratio of p-PI3K/total PI3K and p-AKT/total AKT were increased 6 h after administration of control siRNA with KA into the left hippocampal CA3 region compared with control rats (Figure 7c,d). However, the ratio of p-PI3K/total PI3K and p-AKT/total AKT were reduced in the hippocampal CA3 area 24 h before pre-treated microinjection of siRNA against *pgc-1α* (2 µg) followed by KA microinjection into the hippocampus (Figure 7c,d).

### 2.7. Effect of VEGF/VEGFR2 on Apoptosis and Neuronal Survival in the Hippocampal CA3 Subfield Following Experimental Status Epilepticus

To further confirm the role of PGC-1α/VEGF/VEGFR2 signaling in neuroprotection from hippocampal damage due to prolonged seizure, we investigated the regulatory role of resveratrol on KA-induced hippocampal neuronal cell death. Significantly, pretreatment with resveratrol (100 μmol) attenuated the extent of cleaved caspases-3 expression in the hippocampal CA3 subfield 7 days after KA-induced status epilepticus (Figure 8a). Otherwise, the level of cleaved caspases-3 was significantly increased in the hippocampal CA3 area 24 h before pre-treated microinjection of siRNA against *pgc-1α* (2 μg) followed by KA microinjection into the hippocampus (Figure 8b). Mitigation of neuronal damage by resveratrol (100 μmol) treatment was demonstrated in the hippocampus in both DNA fragmentation qualitative (Figure 9a) and quantitative (Figure 9b) analyses, indexes for cell death, after inducing status epilepticus for 7 days. On the other hand, the level of qualitative (Figure 9c) and quantitative (Figure 9d) analysis of DNA fragmentation were aggravated in the hippocampal CA3 area 24 h before pre-treated microinjection of siRNA against *pgc-1α* (2 μg), followed by KA microinjection into the hippocampus.

## 3. Discussion

The concept of neuroprotection under brain insults, such as prolonged seizures (status epilepticus), should consist of not only the conservation of structural neuronal cell death, but also preservation of neuronal networks and functions [38,39]. In research on prevention of the deleterious effects of status epilepticus, the primary focus is on the development of chronic epilepsy and cognitive decline [38,39]. Thus, a more precise definition of neuroprotection in status epilepticus needs to include protection not just against neuronal death but also against neuronal and network dysfunction at the cellular and molecular levels [39,40]. However, the studies of neuroprotective mechanisms elicited by recurrent seizure activities, especially under status epilepticus, are limited. Considering the detrimental reaction in brain cells under status epilepticus, acute response proteins to counteract these detrimental effects elicited by sustained seizures can be a mechanism of endogenous neuroprotection against seizure-induced neuronal cell death [39,41,42]. Following status epilepticus, the mechanism of endogenous neuronal survival signals is evolutionarily conserved and may actuate extensive signaling pathways to offer the neuroprotective effect which therefore may be strong candidates for therapeutic strategies [39,42]. In our previous studies, both in human epilepsy [43] and animal studies following status epilepticus [5,26,41,44,45,46], several endogenous neuroprotective mechanisms to lessen neuronal damage were proposed, including PPARγ [41], mitochondrial uncoupling protein 2 (UCP2) [41], heat shock protein 70 [43,44], brain-derived neurotrophic factor [45], mitochondrial dynamin-related protein 1 [46], PGC1-α [5,26], and sirtuin 1 [5,26].

The present study demonstrated that activation of PGC-1α activity regulated VEGF/VEGFR2 (Flk-1) signaling and showed the neuroprotective effect against neuronal cell death in the hippocampus following experimental status epilepticus. Particularly, we noted that PGC-1α expression was enhanced in the rat hippocampus following status epilepticus and upregulated VEGF and VEGFR2 (Flk-1) expression. Administration of the activator of PGC-1α, resveratrol, was accompanied by increased VEGF and VEGFR2 expression, and promotion of PI3K/AKT and MEK/ERK pathways and decreased neuronal cell death in the hippocampus following status epilepticus. Downregulation of PGC-1α by pretreatment of siRNA against *pgc-1α* reduced VEGF and VEGFR2 expression and inhibited PI3K/AKT and MEK/ERK pathways, also accompanied by heightened caspase-3 activity, and augmented neuronal damage in the hippocampal CA3 subfield. We also noted that the mRNA levels for *vegf* and *vegfr2* were reduced after pretreatment of siRNA against *pgc-1α* compared with the levels of control siRNA groups. This suggests an additional mechanism for seizure-induced activation following status epilepticus. Therefore, our results indicated that seizure activities triggered the upregulation of the VEGF/VEGFR2 (Flk-1) pathway, activation of PI3K/AKT and MEK/ERK signaling, and worked against the hippocampal neuronal apoptotic cell death following status epilepticus. Therefore, our results showed VEGF/VEGFR2 and related downstream PI3K/AKT and MEK/ERK signaling can exert endogenous neuroprotection in status epilepticus.

Recently, our research revealed that PGC-1α is an important transcriptional regulator that acts through regulating the expression of reactive oxygen species (ROS), mitochondrial UCP2, superoxide dismutase 2, and mitochondrial biogenesis, which plays a beneficial part in neuroprotection in the CNS following global ischemia and status epilepticus [5,26,28,41,47]. In addition, recent studies have demonstrated that the PGC-1α signaling pathway exerts potential neuroprotective properties in many neurological diseases [26,48,49], such as Alzheimer’s disease [50], Parkinson’s disease [51], acute stroke [28,52], epilepsy [53], and status epilepticus [26,53,54,55]. Whether activation of PGC-1α following sustained seizures promotes the endogenous activation of VEGF/VEGFR2 signaling is unclear. However, recent evidence showed activation of PGC-1α induces the expression of VEGF/VEGFR2 signaling, leading to protection from apoptotic cell damage in the neuronal cells and providing neuroprotective effects [15,29,30,31]. Thus, the activation of PGC-1α may regulate the VEGF/VEGFR2 signaling pathway in the neuronal cells and might be a crucial mechanism in neuroprotection following status epilepticus.

VEGFs are strong endothelial cell mitogens and major regulators of neurogenesis and angiogenesis [8,56]. In addition to the well-established effects of VEGF, recent research has demonstrated pivotal roles for VEGF/VEGFR2 in a broad range of neurotrophic and neuroprotective effects in the CNS that may relate to neurogenesis and angiogenesis [11,12,13]. The neurotrophic and neuroprotective effects of VEGF are predominantly mediated by VEGFR2, also called fetal Flk-1 [14,57,58,59], or kinase insert-domain containing receptor (KDR) [60]. VEGF/VEGFR2 (Flk-1) signaling may have neuroprotective effects in many neurological diseases, such as hemorrhagic or ischemic stroke [16,17,18], traumatic brain injury [61], amyotrophic lateral sclerosis [20], Huntington’s disease [62], Alzheimer’s disease [12,21], and Parkinson’s disease [12,22]. Recent evidence showed that during status epilepticus, VEGF is upregulated and it protects against seizure-induced neuronal cell death in the hippocampus [14,19,23]. In vitro, VEGF administration suppresses ictal and interictal epileptiform activity via the VEGFR2 (Flk-1) receptor. Thus, upregulated VEGF signaling through VEGFR2 (Flk-1) overexpression may regulate epileptogenesis and ictogenesis in mice and counteract the focal epileptic seizure [14].

VEGF can directly stimulate the proliferation of neuronal progenitors; however, the possible mechanism of seizure-induced activation of endogenous PGC-1α that promotes the following VEGF signaling that contributes to neuroprotection in the hippocampus following status epilepticus is not clear. Thus, we investigated two downstream pathways of the VEGF/VEGFR2 (Flk-1) signaling pathways that included PI3K/AKT and MEK/ERK signaling [13,63,64,65]. Activation of VEGF has been suggested to regulate PI3K/AKT and MEK/ERK cascades. Furthermore, VEGF is known to be involved in the trophic and neuroprotective effects of growth factors via binding to its tyrosine kinase receptors, particularly of which VEGFR2 (Flk-1) is proposed to mediate most of the neuron-specific effects of VEGF via mediating PI3K/AKT and MEK/ERK signaling [10,11,12,13,33].

Mitogen-activated protein kinases (MAPK) are considered to be a family of Ser/Thr protein kinases and conserved in eukaryotes and involved in many cellular programs. The MEK/ERK pathway is one of the MAPK cascades involved in transducing cell survival signals via growth factor receptors [66,67]. Interestingly, growing evidence revealed that improving mitochondrial function with activation of PI3K/AKT and MEK/ERK pathways might tend to inhibit phosphorylation of c-Jun-NH2-terminal kinase (JNKs) and p38, the second and third major signaling cassettes in the MAPK pathway, which majorly respond to inflammatory and cellular stress to promote inflammation and cell death, in neuronal cells under many neurological conditions, both in vitro and in vivo studies [66,67,68]. Overall, MEK/ERK pathways have been reported to play a crucial role in protecting neuronal cells from death under hypoxia, global ischemia, epilepsy, status epilepticus, and Parkinson’s disease [33,63,68,69]. Therefore, following status epilepticus, endogenous activation of PGC-1α may regulate the VEGF/VEGFR2 (Flk-1) signaling pathway that triggers PI3K/AKT and MEK/ERK cascades and also promotes the cAMP-CREB signaling axis [45,69], which regulates the anti-apoptotic B-cell lymphoma-2 (Bcl-2) family, the expression of autophagy [63] and lysosomal genes, and further contributes to neuron survival mechanisms in hippocampal neuronal cells [70,71,72].

Several studies showed that the PI3K/AKT signaling pathway can modulate cellular activities, such as neuronal proliferation, differentiation, cell survival, and synaptic plasticity, and this signaling cascade activation promotes an important cytoprotective mechanism that promotes neural survival associated with neurodegenerative disorders and epileptic seizures [13,73,74]. Binding of VEGF may further trigger VEGFR2 (Flk-1) to activate PI3K and then phosphorylate phosphatidylinositol 4,5-bisphosphate or PtdIns (4,5)*P*_2_ (PIP_2_) to the second messenger, phosphatidylinositol (3,4,5)-trisphosphate (PtdIns(3,4,5)*P*_3_ (PIP_3_), on the plasma membrane. PIP_3_ directly or indirectly binds to AKT to induce structural changes and facilitate the phosphorylation of AKT amino acid residues under neurodegenerative diseases [13,73,74]. Therefore, under different acute or chronic neurological conditions, activated AKT can promote neuronal cell survival in several ways, including subsequently regulating numerous downstream molecules, such as cyclin D, glycogen synthase kinase-3β, and mechanistic targets of rapamycin complex 1, thereby regulating cell functions and improving cell survival advantage [75]. Moreover, phosphorylated AKT regulates cell survival and anti-apoptosis by targeting the pro- or anti-apoptotic mediators, such as Bcl-2, Bax, and caspases [76,77,78].

In addition to effects of the PI3K/AKT signaling pathway on cellular proliferation and survival, the activation of the MEK/ERK pathway encourages cell survival and prevents cell apoptosis [63,65]. Therefore, the role in neuroprotection of the MEK/ERK pathway is emphasized in many acute or chronic neurological diseases [13,63,79]. Growing evidence has shown that the MEK/ERK pathway plays a central role as a potential therapeutic target in neurological disorder diseases including epilepsy and status epilepticus [13,80,81]. The MEK/ERK pathway is also a vital message transmission pathway in embryonic and adult neurogenesis and plays a role in the regulation of brain physiological function [80]. When the cell suffers external stress, such as prolonged seizures, VEGF may stimulate cellular responses by binding to a special receptor, VEGFR2 (Flk-1), and this activates receptors in the form of the dimer which start signaling output and then triggers the activation of a series of signal transduction molecules, such as MEK/ERK signaling.

Evidence has shown that mitochondrial related autophagic pathways, such as the Jun N-terminal kinases-associated B-cell lymphoma-2 (JNK/Bcl-2) pathway, MEK/ERK signaling, and the sirtuin 1/forkhead box protein O1 pathway, may provide neuroprotective effects from neurotoxicity against dopamine neurons [82]. Activation of the MEK/ERK pathway by phosphorylation of threonine and tyrosine residues can quickly initiate ERK1/2-dependent biological processes in neurological diseases, including epilepsy and status epilepticus [39]. Recently, we noted exogenous resveratrol may provide neuroprotective effects through upregulation of PGC-1α, modulation of mitochondrial dynamics, and MEK/ERK regulated autophagy in rotenone-induced oxidative stress SH-SY5Y cell lines [63]. These results suggest activation PGC-1α regulated the VEGF/VEGFR2 pathway and further promoted MEK/ERK signaling which provided neuroprotective effects against epilepsy, status epilepticus, and neurodegenerative diseases.

## 4. Materials and Methods

### 4.1. Animals

The procedures of experimental status epilepticus in animals were carried out in compliance with the guidelines of Institutional Animal Care and Use Committee (the identification code: 2012122105; approval date was 01 August 2013) and were certificated by the experimental animal ethics committee at Chang Gung Memorial Hospital, Kaohsiung, Taiwan. A sample of 168 pathogen-free adult male Sprague Dawley rats (weight ranged from 280–350 g) were obtained from BioLASCO Taiwan Co. Ltd. (Taipei, Taiwan) and enrolled in the study. The animals were housed in the Center for Laboratory Animals in an environmentally controlled room (12 h/12 h light/dark cycle; 24 ± 1 °C). Tap water and rat chow in the laboratory were available ad libitum. All efforts were made to reduce the number of animals used and to minimize animal suffering during the experiment.

### 4.2. Experimental Status Epilepticus

The animal model of experimental status epilepticus that we used was well established previously [5,26,44]. Briefly, a stereotaxic headholder (Kopf, Tujunga, CA, USA) was used to fix the head of animals after they inhaled 3% of isoflurane for anesthesia, and the body temperature of animals was maintained at 37 °C by heating pads. KA (0.5 nmol; Tocris Cookson, Bristol, UK) dissolved in PBS (0.1 M, pH 7.4) was stereotaxically microinjected into the CA3 left hippocampal subfield (3.3–3.8 mm below the cortical surface, 2.4–2.7 mm from the midline, and 3.2–3.5 mm posterior to bregma) [5,26,44]. Microinjection of KA into the left hippocampal CA3 region caused progressive and accompanying augmented seizure-like hippocampal electroencephalographic (hEEG) activity that was routinely detected from the right hippocampal CA3 [83,84,85]. We therefore observed the seizure activity (right side hEEG) for 60 min, and then the seizure activities were terminated by diazepam (30 mg/kg, i.p.) [5,26]. To avoid post-surgical infection, sodium penicillin (10,000 IU; YF Chemical Corp., New Taipei City, Taiwan) was given intramuscularly. The animals were returned to the recovery room in individual cages. Animals with anesthesia and surgical preparations without additional experimental manipulations served as controls. The experimental schemes as the following Figure 10.

### 4.3. Pharmacological Pretreatments

The test agent included resveratrol (R5010, Sigma-Aldrich, St. Louis, MO, USA) [26,86], a PGC-1α activator, that was microinjected bilaterally and sequentially into the bilateral hippocampal CA3 areas [26]. The dose of resveratrol used was 100 µmol dissolved in 3% dimethyl sulfoxide (DMSO) (D5879, Sigma-Aldrich) as in our previous reports [26,41], at a volume of 150 nL on each side. Rats receiving microinjections of DMSO (3%) were used for the volume and vehicle controls. Each rat received only a single pharmacological pretreatment to prevent the confounding effects of drug–drug interactions 30 min before administration of KA (0.5 nmol) or PBS into the left hippocampal CA3 subfield.

### 4.4. Gene Knockdown by Microinjection of Small Interfering RNA (siRNA) Against pgc-1α into the Hippocampus

Gene knockdown was conducted with small interfering RNA (siRNA) against the *pgc-1α* gene. Multiple siRNA sequences were synthesized form by GE Healthcare (GE Healthcare, Chicago, IL, USA). All siRNAs were dissolved in an isotonic RNAi buffer (100 mM potassium acetate; 30 mM Hepes-KOH; 2 mM magnesium acetate; 26 mM NaCl, pH 7.4, at 37 °C). The siRNA against *pgc-1α* as following blow:
***pgc-1α*****Sequence**Sequence 15′-CGGUGGAUGAAGACGGAUU-3′Sequence 25′-CAAUGAAUGCAGCGGUCUU-3′Sequence 35′-GAACAAGACUAUUGAGCGA -3′Sequence 45′-AUUCAAACUCAGACGAUUU-3′

Pooled siRNAs were mixed with Lipofectin reagent (1:1, 18292-011, Invitrogen, Carlsbad, CA, US) and microinjected bilaterally into the CA3 region of the hippocampus 24 h before the microinjected administration of KA. The pooled non-targeting siRNAs containing four scramble sequences were used as control siRNA.

### 4.5. Collection of Tissue Samples from the Hippocampus

At time-intervals (1, 3, 6, or 24; or 7 days) after microinjection of KA or PBS into the hippocampus, animals were anesthetized with 3% isoflurane and were perfused intracardially [83,85]. The brain was immediately removed and placed on gauze moistened with 0.9% ice-cold saline. To avoid the confounding effect of KA toxicity, we routinely collected samples from the CA3 region of the right hippocampus (this side is the hEEG recording side, with no KA injection). Our method permitted us to verify that our results from the analyzed samples were directly from sustained seizures (status epilepticus) and not indirectly due to KA toxicity [5,26,44]. Hippocampal samples were stored at −80 °C until use in biochemical analyses.

### 4.6. RNA Isolation and Reverse Transcription Real-Time Polymerase Chain Reaction

To quantitatively analyze the expression of *pgc-1α*, *vegf*, and *vegfr2 (Flk-1)* mRNA in the tissue of the hippocampal CA3, the brain was rapidly removed after PBS perfusion and total RNA from the hippocampus was extracted with a RNeasy Mini Kit (Qiagen, Duesseldorf, Germany) according to the manufacturer’s protocol as in our previous reports [5,26,44]. Reverse transcriptase (RT) reaction was applied with the ImProm II^TM^ Reverse Transcription System (Promega, Madison, WI, USA) to acquire first-strand cDNA synthesis. Real-time polymerase chain reaction (PCR) for amplification of cDNA was performed using a LightCycler^®^ 480 SYBR Green I Master (Roche Diagnostics, Mannheim, Germany). The qPCR of each sample was carried out in duplicate for all cDNA and for glyceraldehyde-3-phosphate dehydrogenase (GAPDH), the housekeeping gene used as a control [5,26,44]. The primer pairs for amplification of *pgc-1α, vegf*, and *vegfr2 (Flk-1)*, and *Gapdh* cDNA used in this study were as follows:
***Gene*****Forward Primer****Reverse Primer***pgc-1α*5′-GTTTCATTACCTACCGTTACAC-3′5′-ATCGTCTGAGTTTGAATCTAGG-3′*vegf*5′-GCAGATGTGAATGCAGACCA-3′5′-TTTCCCTTTCCTCGAACTGA-3′*vegfr2*5′-AAGCAAATGCTCAGCAGGAT-3′5′-GAGGTAGGCAGGGAGAGTCC-3′*Gapdh*5′-AACGGCACAGTCAAGGCTGA-3′5′- ACGCCAGTAGACTCCACGACAT -3′

The products of PCR were subsequently subjected to agarose gel electrophoresis for further confirmation of amplification specificity [5,26,44]. Fluorescence signals from the amplified products were quantitatively assessed using the LightCycler software program (version 3.5). The second derivative maximum mode was chosen with baseline adjustment set in the arithmetic mode. The relative change in target mRNA expression was determined by the fold-change analysis [5,26,44], in which fold change = 2^−^[ΔΔC*t*], where ΔΔC*t* = (C*t*,_target mRNA_ − C*t*,_GAPDH_).

### 4.7. Western Blotting Analysis

Western blot analysis was carried out on proteins extracted from total lysate or cytosolic fractions from hippocampal samples. The primary antiserum used included rabbit monoclonal or polyclonal antibodies against PGC-1α (sc-13067, Santa Cruz Biotechnology, Heidelberg, Germany), VEGF (5365-100, BioVision Huissen, Netherlands), VEGFR2 (9698S, Cell signaling, Danvers, MA, USA), PI3 Kinase (4257S, Cell signaling), Phospho-PI3 Kinase (4228S, Cell signaling), AKT (9272, Cell signaling), Phospho-AKT (4060S, Cell signaling), ERK1/2 (9102S, Cell signaling), Phospho-ERK (4370S, Cell signaling), Cleaved caspase-3 (9664, Cell signaling), and β-actin (ab8227, abcam, Cambridge, UK). The primary antibodies were diluted in 5% skimmed milk in TBST. The conditions for the membrane wash, detection of immunoreactive signals, and quantification of signal intensities on the blots were performed as previously described [5,26,44,46]. This was followed by incubation with horseradish peroxidase-conjugated secondary goat anti-rabbit IgG (111-035-045, Jackson ImmunoResearch, West Grove, PA, USA) to detect the first antibodies for PGC-1α, VEGF, VEGFR2, PI3 Kinase, Phospho-PI3 Kinase, AKT, Phospho-Akt, ERK1/2, Phospho-ERK Cleaved caspase-3, and β-actin. Specific antibody–antigen complexes were detected by an enhanced chemiluminescence western HRP substrate (Merck Millipore, Billerica, MA, USA). ImageJ software (National Institutes of Health, Bethesda, MD, USA) was applied to quantify the amount of proteins and expressed as the ratio relative to β-actin protein.

### 4.8. Double Immunofluorescence Staining and Laser Confocal Microscopy

Double immunofluorescence staining [5,26,44,46,84] was carried out using a goat polyclonal antiserum against VEGF (Santa Cruz Biotechnology) and a mouse monoclonal antiserum against a specific marker for neurons, neuron-specific nuclear protein (NeuN) (Chemicon, Temecula, CA, USA). A goat anti-mouse IgG conjugated with Alexa Fluor 568 (Molecular Probes, Eugene, OR, USA) and a goat anti-rabbit IgG conjugated with AlexaFluor 488 (Molecular Probes) served as the secondary antisera. The sections of right hippocampal CA3 tissue were inspected under an epifluorescence microscope (Olympus AX-51; Olympus, Kyoto, Japan). Under the epifluorescence microscope, the immunoreactivity for VEGF exhibited green fluorescence, and NeuN manifested red fluorescence.

### 4.9. Qualitative and Quantitative Analysis of DNA Fragmentation

Hippocampal CA3 tissues were subject to measurement of the apoptotic DNA fragmentation following status epilepticus [5,26,44,85]. After total DNA was extracted from the hippocampal CA3 tissues, nucleosomal DNA ladders were amplified by a PCR kit for DNA ladder assay (Cat.#: APO-DNA1; Maxim Biotech, San Francisco, CA, USA) to heighten the sensitivity according to the manufacturer’s protocol [5,26,44,85]. For quantification of apoptotic DNA fragmentation, a cell death detection ELISA kit (Cat.#11774425001; Roche Molecular Biochemicals) was performed to detect the cytosolic level of histone-associated DNA fragments according to the manufacturer’s protocol [5,26,44,85]. Proteins from hippocampal tissues served as the source of antigen, jointly with primary anti-histone antibody and secondary anti-DNA antibody coupled with horseradish peroxidase. The cytosolic nucleosomes were quantitatively measured using 2,2′-azino-di-[3-ethylbenzthiazoline] sulfonate as a substrate. Absorbance was determined at 405 nm and referenced at 490 nm using a Multiskan Spectrum reader (Thermo Scientific, Miami, OK, USA).

### 4.10. Statistical Analysis

The continuous variables were expressed as mean ± standard error of the mean (SEM). The statistical method was used to compare all the continuous variables with one-way analysis of variance (one-way ANOVA). Once the ANOVA was significant, for post hoc assessment, we used Scheffé multiple range test to assess the difference between groups, especially the experimental group and the control group. A *p* < 0.05 was considered statistically significant.

## 5. Conclusions

In the present study, our results suggested activation of VEGF/VEGFR2 (Flk-1) by PGC-1α subsequently activates PI3K/AKT and MEK/ERK signaling by alternating the activation state of numerous of downstream proteins relevant to cell proliferation and inhibiting apoptosis in the hippocampus following status epilepticus. Otherwise, downregulation of PGC-1α by siRNA against to *pgc-1α* will inhibit VEGF/VEGFR2 (Flk-1) signaling and suppress pro-survival PI3K/AKT and MEK/ERK pathways that also are accompanied by hippocampal CA3 neuronal cell apoptosis. These results may indicate that the PGC-1α induced VEGF/VEGFR2 pathway may trigger neuronal survival signaling and PI3K/AKT and MEK/ERK signaling pathways. Thus, the axis of PGC-1α/VEGF/VEGFR2 (Flk-1) and downstream PI3K/AKT and MEK/ERK signaling could be considered as an endogenous neuroprotective effect against apoptosis in the hippocampus following status epilepticus. Taking together, our previous research and the results from this study, new strategies for treatment, disease modification, and neuroprotection in epilepsy, particularly in status epilepticus, should be developed.

## Figures and Tables

**Figure 1 ijms-21-07247-f001:**
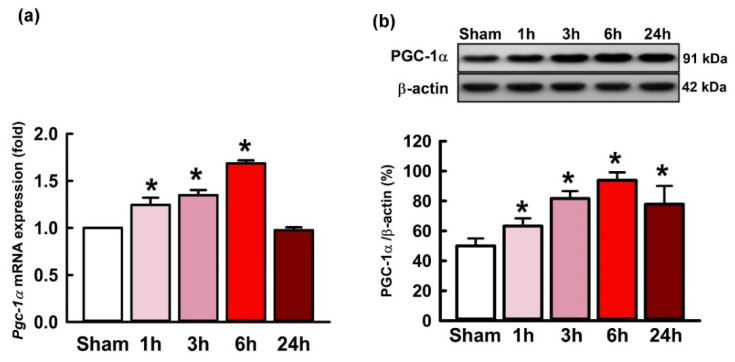
(**a**) Changes of the *pgc-1α* mRNA expression and (**b**) the protein levels of PGC-1α after kainic acid (KA; 0.5 nmol) was microinjected into the hippocampal CA3 subfield. Tissues were harvested from the right subfield CA3 of the hippocampus at 1, 3, 6, or 24 h after administration of KA or phosphate buffered saline (PBS) into the left hippocampal CA3. Values are presented as mean ± SEM of four animals per experimental group. * *p* < 0.05 versus the control group in the Scheffé multiple-range test.

**Figure 2 ijms-21-07247-f002:**
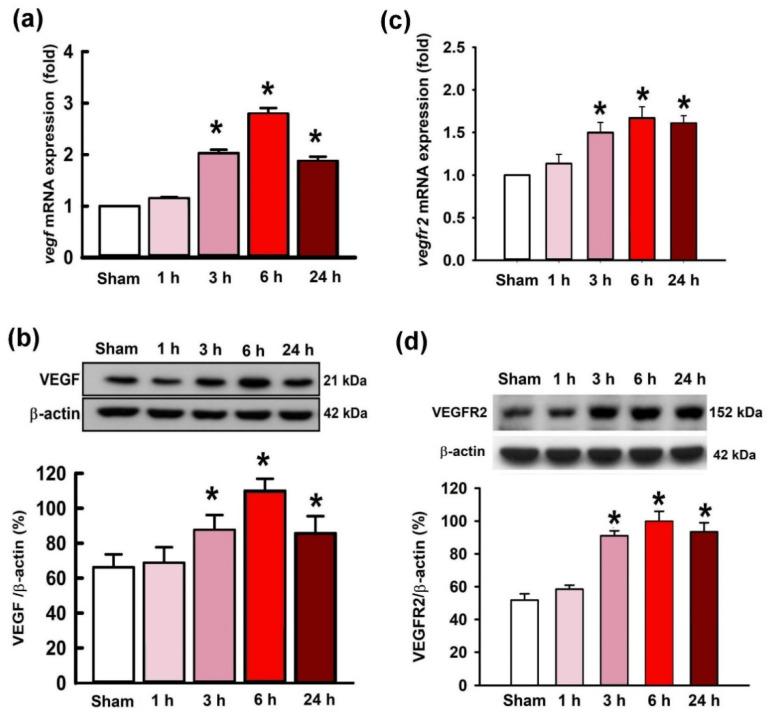
Upregulated changes of the expression of vascular endothelial growth factor (*vegf*) (**a**) and VEGF receptor 2 (*vegfr2*) (**c**) mRNA levels, and VEGF (**b**) and VEGF receptor 2 (VEGFR2) protein levels (**d**) after the administration of KA (0.5 nmol) into the hippocampal CA3 region. Samples were collected from the right hippocampal CA3 subfield at 1, 3, 6, or 24 h after the administration of KA or PBS into the left hippocampal CA3 subfield. Values are presented as mean ± SEM of quadruplicate analyses from four animals per experimental group. * *p* < 0.05 versus the control group in the Scheffé multiple-range test.

**Figure 3 ijms-21-07247-f003:**
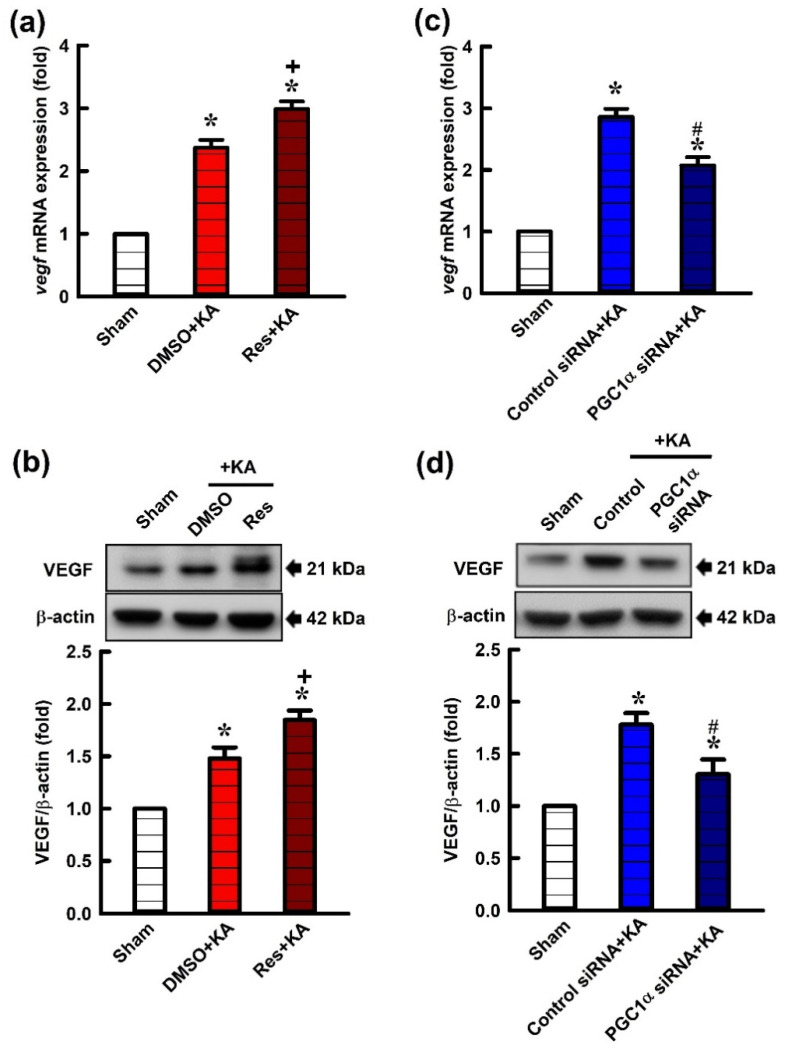
(**a**) Changes of *vegf* mRNA expression and (**b**) VEGF protein levels relative to β-actin from the CA3 of the hippocampus 6 h after microinjection of 0.5 nmol kainic acid (KA) or pretreatment with microinjection of resveratrol (Res; 100 μmol) or 3% dimethyl sulfoxide (DMSO) into the hippocampal CA3 subfield. (**c**) Changes of *vegf* mRNA expression in the CA3 of the hippocampus 6 h after the microinjection of KA (0.5 nmol) into the left hippocampal CA3 subfield, and 24 h before pre-treatment with application into the bilateral CA3 subfield of control siRNA or siRNA for *pgc-1α* (2 μg). (**d**) Changes in VEGF relative to β-actin from the CA3 subfield of the hippocampus 6 h after the same experiments. Values are mean ± SEM of quadruplicate analyses from 6–8 animals per experimental group. * *p* < 0.05 versus the control group, + *p* < 0.05 versus DMSO+KA group, and # *p* < 0.05 versus control siRNA+KA group in the Scheffé multiple-range test.

**Figure 4 ijms-21-07247-f004:**
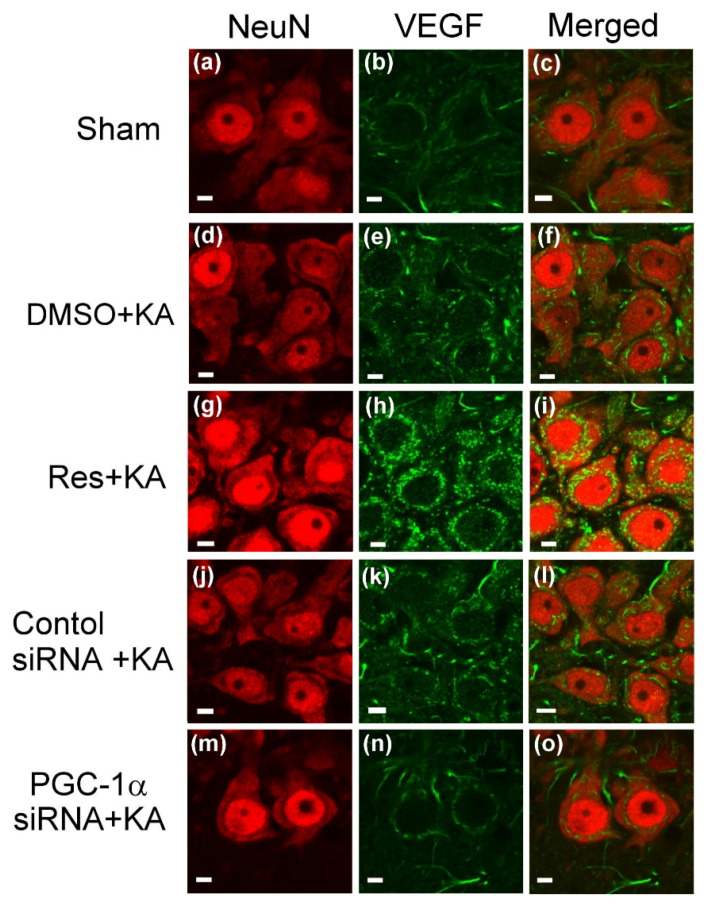
Confocal microscopic images of the right CA3b subregion of the hippocampus showing cells that were immunoreactive to neuron-specific nuclear protein (NeuN) (red fluorescence) or additionally stained for VEGF (green fluorescence) in control rats (Figure 4a–c), or 6 h after administration of kainic acid (KA; 0.5 nmol) (Figure 4d–f) or pre-treatment with resveratrol (Res; 100 μmol) 30 min before KA-induced status epilepticus (Figure 4g–i). Induction of status epilepticus by KA in animals that received pre-treatment with application into the bilateral CA3 subfield 24 h after animals received control siRNA (Figure 4j,i) or siRNA against *pgc-1α* (2 μg) (Figure 4m–o). Scale bar, 10 μm.

**Figure 5 ijms-21-07247-f005:**
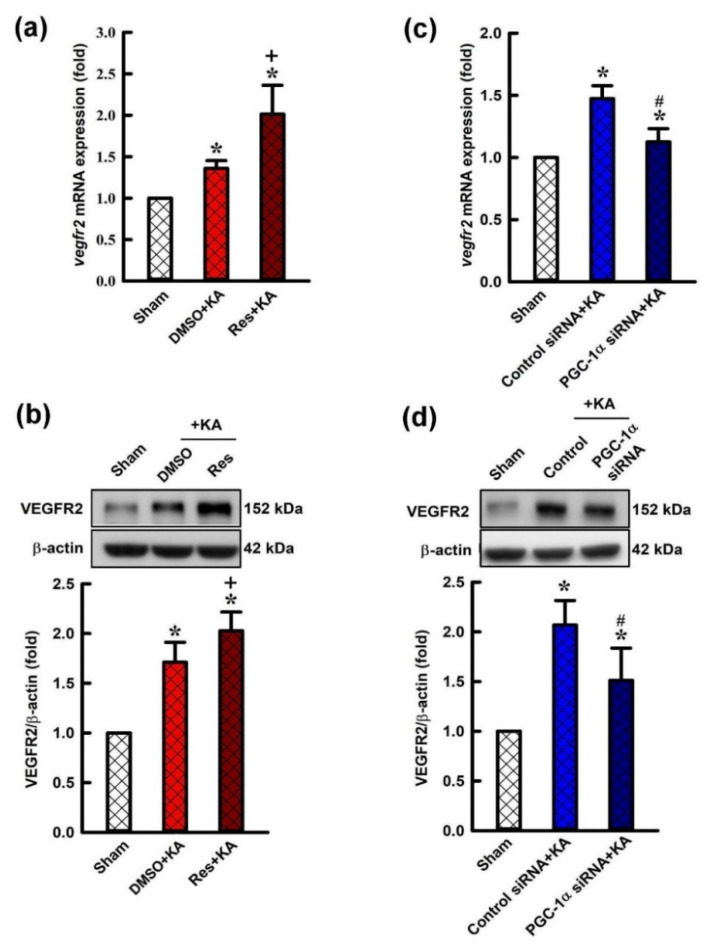
(**a**) Changes of *vegfr2* mRNA expression and (**b**) changes of VEGFR2 protein in the hippocampal CA3 tissues 6 h after administration with kainic acid (KA; 0.5 nmol) with pre-treatment with resveratrol (Res; 100 μmol), or 3% DMSO into bilateral CA3 subfield. (**c**) The *vegfr2* mRNA expression in the hippocampus 6 h after the administration of KA into the left CA3 area 24 h after pre-treatment of control siRNA or *pgc-1α* siRNA (2 μg) into the bilateral CA3 subfield. (**d**) Changes in VEGFR2 protein expression in the CA3 tissue 6 h after the same treatments. Values are presented as mean ± SEM from four animals per experimental group. * *p* < 0.05 versus control groups, + *p* < 0.05 versus DMSO+KA group, and # *p* < 0.05 versus control siRNA+KA group in the Scheffé multiple-range test.

**Figure 6 ijms-21-07247-f006:**
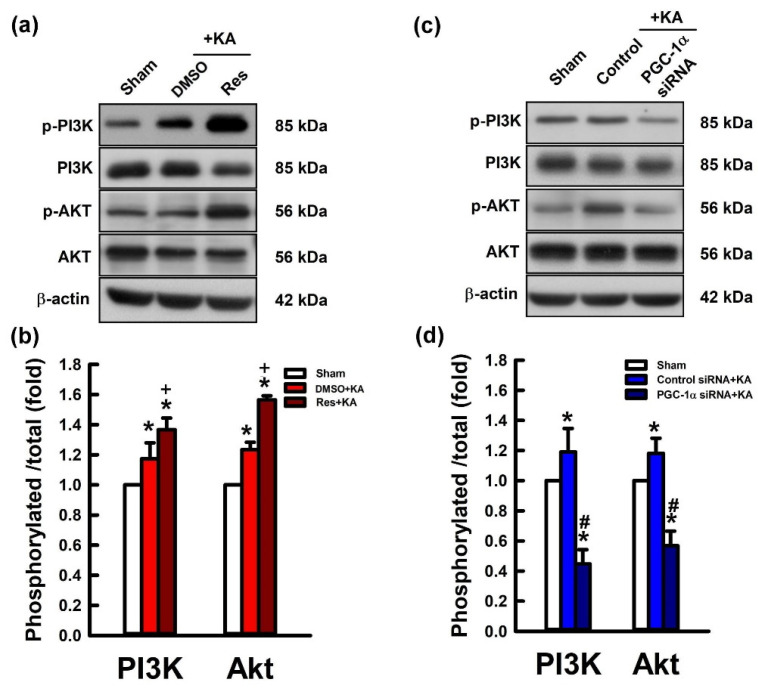
(**a**) Changes of phosphorylated phosphatidylinositol 3-kinase (p-PI3K), total PI3K, phosphorylated AKT (p-AKT), and total AKT expression and (**b**) p-PI3K protein levels relative to total PI3K and p-AKT relative to total AKT in the CA3 region of the hippocampus 6 h after microinjection of kainic acid (KA; 0.5 nmol) with pretreatment of microinjection of 3% DMSO or resveratrol (Res; 100 μmol) into the hippocampal CA3 subfield. (**c**) Ratio of activated phosphorylation of PI3K and AKT levels in the CA3 subfield of the hippocampus 6 h after the application of KA (0.5 nmol) into the left side of the hippocampal CA3 region 24 h before treatment with control siRNA or siRNA for *pgc-1α* (2 μg) application into the bilateral CA3 subfield. (**d**) Phosphorylation levels of PI3K and AKT in the CA3 region of the hippocampus after 6 h of the same experiments. Values presented as mean ± SEM from six animals per experimental group. * *p* < 0.05 versus the control group, + *p* < 0.05 versus DMSO+KA group, and # *p* < 0.05 versus control siRNA+KA group in the Scheffé multiple-range test.

**Figure 7 ijms-21-07247-f007:**
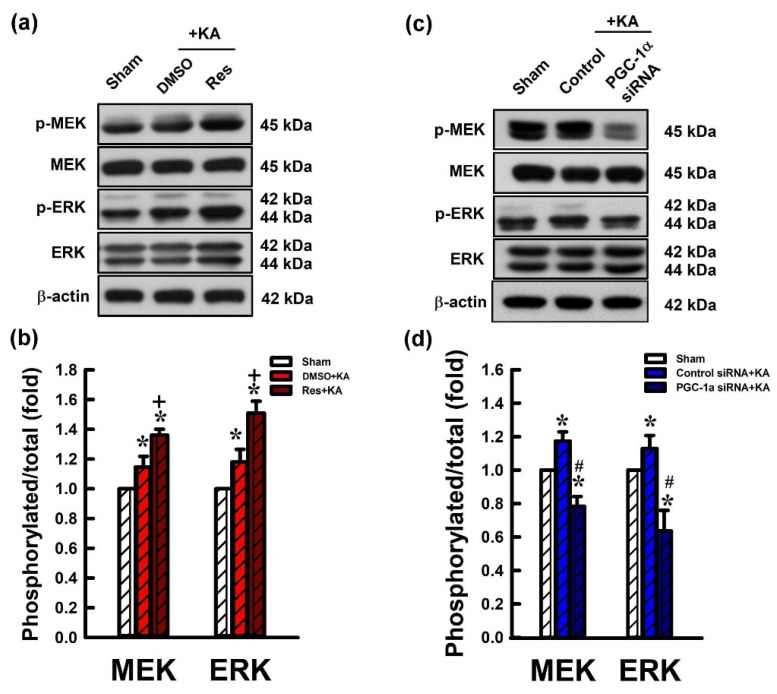
(**a**) Changes of phosphorylated mitogen activated protein kinase (MEK) (p-MEK), total MEK, phosphorylated MEK (p-ERK), and total ERK expression, and (**b**) ratio of phosphorylation level for MEK protein levels and total MEK and phosphorylation of ERK in the CA3 tissue of the hippocampus 6 h after application of kainic acid (KA; 0.5 nmol) with pre-treated microinjection of 3% DMSO or resveratrol (Res; 100 μmol) into the hippocampal CA3 subfield. (**c**) Changes of p-MEK, total MEK, p-ERK, and total ERK expression in the hippocampus CA3 region 6 h after the microinjection of KA (0.5 nmol) into the left CA3 region with pre-treatment of application into the bilateral CA3 subfield with control siRNA or siRNA for *pgc-1α* (2 μg) 24 h in advance. (**d**) Ratio of phosphorylation level for MEK protein levels and total MEK and phosphorylation of ERK in the CA3 region of the hippocampus 6 h after the same experiments. Values are mean ± SEM of quadruplicate analyses from six animals per experimental group. * *p* < 0.05 versus control group, + *p* < 0.05 versus DMSO+KA group, and # *p* < 0.05 versus control siRNA+KA group in the Scheffé multiple-range test.

**Figure 8 ijms-21-07247-f008:**
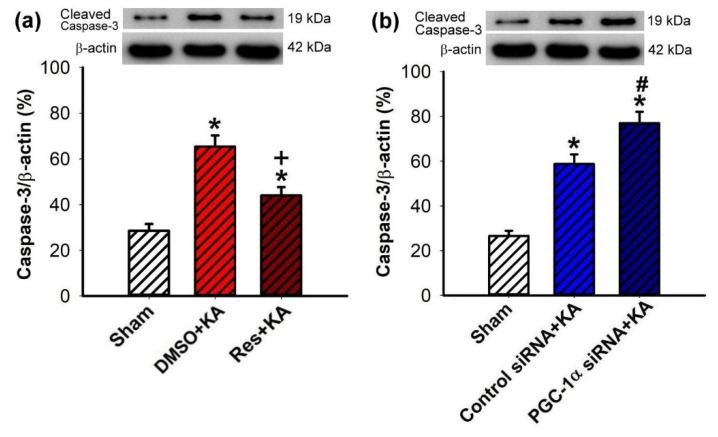
(**a**) Changes of cleaved caspase-3 levels 7 days after microinjection of 0.5 nmol of kainic acid (KA) with 24-h pretreatment with resveratrol (Res; 100 μmol) or 3% DMSO into the hippocampal CA3 subfield. (**b**) Increase of cleaved caspase-3 (19 kDa) 7 days after rats were microinjected in the bilateral CA3 subfields with control siRNA or siRNA for *pgc-1α* (2 μg) 24 h before the microinjection of KA into the left hippocampal CA3. Values are mean ± SEM of quadruplicate analyses from four animals per experimental group. * *p* < 0.05 versus control group, + *p* < 0.05 versus DMSO+KA group, and # *p* < 0.05 versus control siRNA+KA group in the Scheffé multiple-range test.

**Figure 9 ijms-21-07247-f009:**
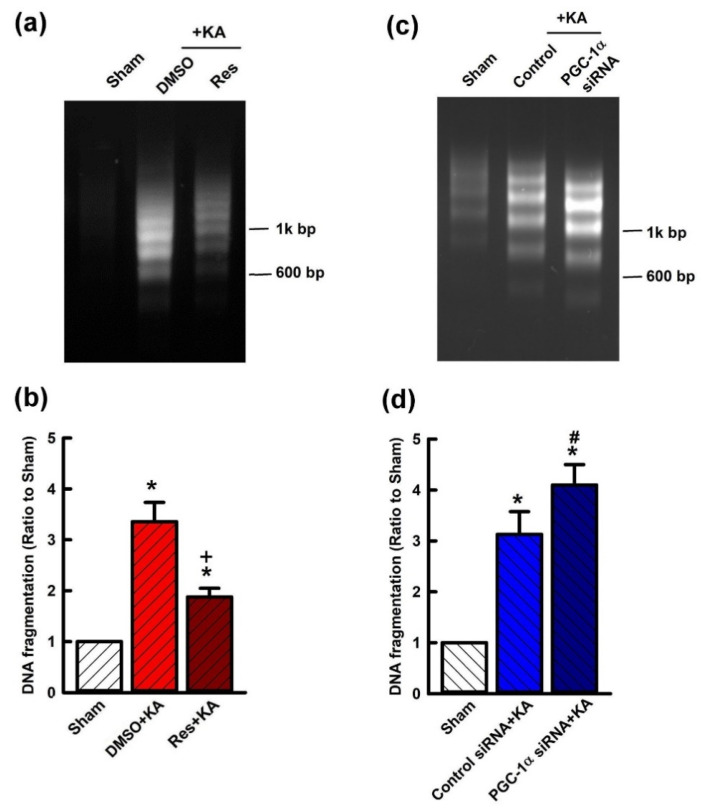
Qualitative (**a**),(**c**) or quantitative (**b**),(**d**) analysis of DNA fragmentation detected in samples harvested from the hippocampus 7 days after status epilepticus with or without Res and *pgc-1α* siRNA. Values are mean ± SEM from six animals per experimental group. (**a**),(**b**) Seven days after microinjection of 0.5 nmol kainic acid (KA) with 24-h pretreatment of microinjection of resveratrol (Res; 100 μmol) into the hippocampal CA3 subfield decreased fragmentation of DNA. Changes of qualitative (**c**) or quantitative (**d**) DNA fragmentation 7 days after microinjected control siRNA or siRNA for *pgc-1α* (2 μg) into bilateral CA3 subfields with 24 h before the microinjection of KA into the left hippocampal CA3. Values are mean ± SEM of quadruplicate analyses from four animals per experimental group. * *p* < 0.05 versus control group, + *p* < 0.05 versus DMSO+KA group, and # *p* < 0.05 versus control siRNA+KA group in the Scheffé multiple-range test.

**Figure 10 ijms-21-07247-f010:**
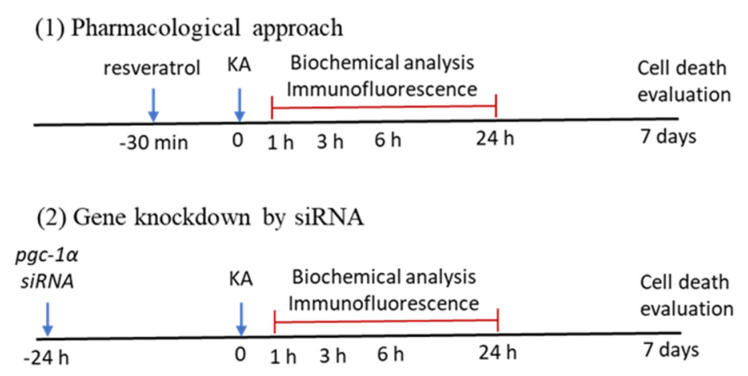
Two experimental schemes which included pharmacological pretreatment of resveratrol and gene knockdown by siRNA against *pgc-1α* following status epilepticus.

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
