# Peer review of "Peroxisome Proliferator-Activated Receptor γ Coactivator 1α Activates Vascular Endothelial Growth Factor That Protects Against Neuronal Cell Death Following Status Epilepticus through PI3K/AKT and MEK/ERK Signaling"

_ijms, 2020, doi:10.3390/ijms21197247_

Round 1

Reviewer 1 Report

The study builds on previous work (both by the authors and others) that implicated Pgc-1α and VEGF signalling in the response to seizures. The novelty of the study is to demonstrate the link between Pgc-1α and VEGF-A/VEGFR-2 expression and activation in the hippocampus following the experimental induction of epilepsy. The experimental design is sound, and the data are well presented and convincing. The conclusions are based on the experimental evidence, and are in line with previous studies.

The use of English needs thorough review. While in most cases it is possible to understand what the authors try to say, there are a large number of grammatical and word use problems that affect the readability and accuracy of the text. Only some of issues are listed in the detailed comments below, so the whole text (and most notably the Materials and Methods section) need to be reviewed for correct language.

Specific questions and comments:

Knockdown of Pgc-1α: How was the efficiency and specificity of the siRNA tested? It would be good to see the Pgc-1α protein levels in the siRNA-treated samples.

Fig. 3c/d: Interesting that Vegfa mRNA levels remain above the control, but the protein levels drop below. Any suggestions? Is this consistent with the immunofluorescence results (Fig. 4)? This should be discussed.

Fig. 3c/5c: mRNA levels for Vegfa and Vegfr2 are reduced after Pgc-1α, but not to control levels. This suggests an additional mechanism for seizure-induced activation. Again, this should be discussed.

Section 4.6: Which qPCR reagents (ie master mix) were used?

Section 4.10: Since only the pairwise comparisons with the sham control are evaluated, is there a specific reason why the authors used Scheffé's method instead of the Tukey method?

Detailed suggestions:

Line 68: Need to specify which VEGF(s) and which VEGFR(s) this line and the manuscript in general refers to. There is a bit of clarification later on in the paragraph. If the authors specifically mean VEGF-A and VEGFR-2, this should be stated at the start.

Line 80: ‘Thus’ is misleading, since the preceding sentence does not provide a causal link. Delete the word.

Lines 88-90: This sentence contradicts the statement in lines 80-82 and ignores references 14 and 19. Do you mean to say that the mechanism of VEGF-A/VEGFR-2 activation after seizure is unclear?

Line 98: Activation should not be capitalised.

Lines 105-107: The sentence does not make sense. Needs re-phrasing.

Lines 107-110: Also not fully clear. Please re-phrase.

Line 110: Do you mean ‘Growing evidence’?

Line 114: MEK is the mitogen activated protein kinase kinase. ERK is the mitogen activated protein kinase.

Line 155: Delete ‘active’

Line 158: Replace ‘tended to test’ with ‘tested’

Line 178: DMSO (not DSMO)

Line 225: Activation of VEGF/VEGFR2

Line 231: delete ‘for’

Author Response

Response to Reviewer#1

  1. Knockdown of Pgc-1α: How was the efficiency and specificity of the siRNA tested? It would be good to see the Pgc-1α protein levels in the siRNA-treated samples.

Response: Using both western blotting analysis and immunohistochemistry. We showed the specificity of siRNA against for pgc-1α. The western blotting analysis showed that administration of decreased siRNA against for pgc-1α decrease the protein level of Pgc-1α (Figure a). Moreover, pretreatment of Pgc-1α siRNA decreased PGC-1α immunoreactivity (Figure b).

  1. 3c/d: Interesting that Vegfa mRNA levels remain above the control, but the protein levels drop below. Any suggestions? Is this consistent with the immunofluorescence results (Fig. 4)? This should be discussed.

Response: In the Figure 3c/d, vegf mRNA level is not consistent with VGGF protein levels (in the results of siRNA against for pgc-1α+KA group) may be related to lower animal numbers. Because of the ongoing project to extend this study, we have extent samples in the same treatments of this study to re-examine the vegf mRNA levels and protein expressions. Thus, we increased the sample size to 6-8 animals in each group and the results was presented in new Figure 3.

  1. 3c/5c: mRNA levels for Vegfa and Vegfr2 are reduced after Pgc-1α, but not to control levels. This suggests an additional mechanism for seizure-induced activation. Again, this should be discussed.

Response: Thank you for your suggestion we have elaborated our “DISCUSSION” (P.350-352).

  1. Section 4.6: Which qPCR reagents (ie master mix) were used?

Response: Real-time polymerase chain reaction (PCR) for amplification of cDNA was performed using a LightCycler® 480 SYBR Green I Master (Roche Diagnostics, Mannheim, Germany).(P.509).

  1. Section 4.10: Since only the pairwise comparisons with the sham control are evaluated, is there a specific reason why the authors used Scheffé's method instead of the Tukey method?

Response: The continuous variables were expressed as mean ± standard error of the mean (SEM). The statistical method was used to compare all the continuous variables with one-way analysis of variance (one-way ANOVA). Once the ANOVA was significant, for post hoc assessment, we used Scheffé multiple range test to assess the difference between groups, especially, the experimental group and the sham control group. A p < 0.05 was considered statistically significant.

 Our manuscript have been submitted to MDPI for English editing: English ID: english-22526

 Detailed suggestions:

 Line 68: Need to specify which VEGF(s) and which VEGFR(s) this line and the manuscript in general refers to. There is a bit of clarification later on in the paragraph. If the authors specifically mean VEGF-A and VEGFR-2, this should be stated at the start.

Response: Thank you for your comments. We have re-wrote and elaborated the “INTRODUCTION” of VEGF-A and VEGFR-2 (P. 68-77).

Line 80: ‘Thus’ is misleading, since the preceding sentence does not provide a causal link. Delete the word.

Response: We have delete “Thus”.

Lines 88-90: This sentence contradicts the statement in lines 80-82 and ignores references 14 and 19. Do you mean to say that the mechanism of VEGF-A/VEGFR-2 activation after seizure is unclear?

Response: Whereas the neuroprotective role of VEGF in ischemic stroke has been well studied, in epileptic seizure, such as status epilepticus, the literatures in neuroprotective role of VEGF-A/VEGFR2 pathway is related limited. (P.90-92)

Line 98: Activation should not be capitalised.

Response: We have corrected it.

Lines 105-107: The sentence does not make sense. Needs re-phrasing.

Response: Thank you your suggestion. The sentence is not clear. We have elaborated the discussion (P.108-114).

Lines 107-110: Also not fully clear. Please re-phrase.

Response: We have elaborated the discussion (P.108-114).

Line 110: Do you mean ‘Growing evidence’?

Response: We have changed “Growing evidence”

Line 114: MEK is the mitogen activated protein kinase kinase. ERK is the mitogen activated protein kinase.

Response: We have corrected it

Line 155: Delete ‘active’

Response: We have deleted it.

 Line 158: Replace ‘tended to test’ with ‘tested’

Response: We have corrected it

 Line 178: DMSO (not DSMO)

Response: We have corrected it

 Line 225: Activation of VEGF/VEGFR2

Response: We have corrected it

Line 231: delete ‘for’

Response: We have deleted it.

Reviewer 2 Report

In this paper, the authors present a series of experiments aiming to demonstrate that PGC-1α activation can exert a neuroprotective effect, quantified as reduction of hippocampal neuronal cell death, through the activation of PI3K/AKt and MEK/ERK pathways. Further, they propose that resveratrol, a molecule well-known for its various beneficial effects, might act as an activator of the aforementioned endogenous pathways, thus boosting an innate protective response after status epilepticus.

The finding itself is interesting and the authors managed to produce a significant amount of data to prove their hypothesis. Nonetheless, there are some points which might deserve further consideration. In particular:

MAJOR POINTS:

1)The authors are proposing a to enhance an endogenous protective mechanism via activation of the downstream pathways of the VEGF/VEGFR2 system. It was found that resveratrol, at the concentration of 100 micromolar, can boost the vegf mRNA expression which would lead to a protective hyperactivation of MEK/ERK and PI3K signaling. This finding is interesting but some issues need further consideration:  

  • Was the effect dose dependent? is it possible that the reported effect is not the plateau?
  • Even if the results are convincing, the link between resveratrol and status epilepticus is weak. The authors justify their findings by referring to the potential reported neuroprotective effects of resveratrol, but none of them (regulation of ROS production, mitochondrial UCP2. Superoxide dismutase…and so on) is investigated in the present work from a functional point of view. This diminishes the potential of the paper since the results confirm the potential role of this substance in status epilepticus without giving further insight in its mechanism of action.
  • Did the authors observe any difference in the onset of status epilepticus comparing Resveratrol treated mice with KA-only treated mice?

2) The quality of the english is overall poor and needs to be improved. Only some examples follow:

Some examples:

Line 62: “significantly” should read “significant”
Line 67: “contributes” should read “contribute”.
Line 155: “it can active activate…” is a repetition
Figure 3, legend: should “DSMO” read “DMSO” (Dimethyl Sulfoxide)? Otherwise clarify this abbreviation.
Line 450-452: The sentence starting with “decreased” should be rephrased.
Line 322: “the conception of neuroprotection” should read (?) “the concept of neuroprotection”

Minor points:

Methods:

  1. The define that “sham” animals only underwent anesthesia and surgical procedures. By this definition, it appears that the injection of the vehicle (DMSO) was not performed in the sham mice. However, in paragraph 4.3 it is stated that some rats received only DMSO microinjections and were used as controls.This is an important point since the concentration of this solvent (3%) is rather high and it could, by itself, exert some toxic effect in mice. Can the authors give a feedback on this issue?
  2. To this reviewer’s comprehension each set of experiments was performed on different groups of rats sacrificed at different time points. Since the number of experiments is considerable, a schematization or a table showing how exactly the total number of animals used in this study is subdivided among the various experiments is needed to better understand the study design.

Figures:

It would be advisable to use a coherent color pattern throughout the whole manuscript. Here, some figures have colors while others are black and white. Please select a single style and use it for all the illustrations.

Figure 4: please change “Contol” to “Control”

Title: The title is somehow complicated. A simpler title would be easier to comprehend for the readers.

Author Response

Response to Reviewer 2#

The authors are proposing a to enhance an endogenous protective mechanism via activation of the downstream pathways of the VEGF/VEGFR2 system. It was found that resveratrol, at the concentration of 100 micromolar, can boost the vegf mRNA expression which would lead to a protective hyperactivation of MEK/ERK and PI3K signaling. This finding is interesting but some issues need further consideration: 

  1. Was the effect dose dependent? is it possible that the reported effect is not the plateau?

Response:

Determination of dosage resveratrol (the response of resveratrol showed dose-dependent)

In some animal without induced status epilepticus, bilateral microinjection of resveratrol (50, 100, or 200 mmol) into hippocampal CA3 region, significantly increased the expression of PGC-1α in the CA3 subfield 6 h (PGC-1α), the effects showed a dose dependent with maximum effect of 100 mmol. Thus, the dose of resveratrol used in the following studies for the purpose to test the effect of resveratrol was 100 mmol.

  1. Even if the results are convincing, the link between resveratrol and status epilepticus is weak. The authors justify their findings by referring to the potential reported neuroprotective effects of resveratrol, but none of them (regulation of ROS production, mitochondrial UCP2. Superoxide dismutase…and so on) is investigated in the present work from a functional point of view. This diminishes the potential of the paper since the results confirm the potential role of this substance in status epilepticus without giving further insight in its mechanism of action.

Response: In our previous studies, both in human [43] and animal studies [5,26,41,44-46], we have found several endogenous neuroprotective mechanisms to lessen neuronal damage following status epilepticus or epilepsy, including PPARγ [41], mitochondrial uncoupling protein 2 (UCP2) [41], heat shock protein 70 [43,44], brain-derived neurotrophic factor [45], mitochondrial dynamin-related protein 1 [46], PGC1-α [5,26] and sirtuin 1 (SIRT1) [5,26]. (Please see Line 339-344).

In the present study, we used pharmacological intervention and gene-knockdown (PGC-1α). Therefore, we can potentially suggested the VEGF/VEGFR-2 pathway is an independently endogenous neuroprotective pathways.

  1. Did the authors observe any difference in the onset of status epilepticus comparing Resveratrol treated mice with KA-only treated mice?

Response:

We did not find resveratrol affects the sustained seizure activity during hEEG monitoring. Also, the behavior of animal is not different in KA-induced experimental temporal lobe status epilepticus. Thus, we suggest the resveratrol have neuroprotective effect that is independent to ictal seizures in aimals.

Some examples:

Line 62: “significantly” should read “significant”

Line 67: “contributes” should read “contribute”.

Line 155: “it can active activate…” is a repetition

Figure 3, legend: should “DSMO” read “DMSO” (Dimethyl Sulfoxide)? Otherwise clarify this abbreviation.

Response: These mistakes have been corrected.

Line 450-452: The sentence starting with “decreased” should be rephrased.

Response: We changed” All efforts were made to reduce the number of animals used and to minimize animal suffering during the experiment.”

Line 322: “the conception of neuroprotection” should read (?) “the concept of neuroprotection”

Response: This word has been changed

Our manuscript have been submitted to MDPI for English editing: English ID: english-22526

Minor points:

Methods:

  1. The define that “sham” animals only underwent anesthesia and surgical procedures. By this definition, it appears that the injection of the vehicle (DMSO) was not performed in the sham mice. However, in paragraph 4.3 it is stated that some rats received only DMSO microinjections and were used as controls. This is an important point since the concentration of this solvent (3%) is rather high and it could, by itself, exert some toxic effect in mice. Can the authors give a feedback on this issue?

Response: In our experience, the target dose of resveratrol can only be solved in 3% DMSO. If we used the lower concentration of DMSO, the resveratrol can not be totally solved to our target dosage (at a volume of 150 nL). Moreover, we tried the 3% DMSO can not influence the cellular and molecular events in the experiences, even in the previous, we tested the mitochondrial respiratory function. 3% DMSO did not showed statistically significant compared PBS. However, high concentration of DMSO showed toxic effect and depressed mitochondrial respiratory chain functions.

  1. To this reviewer’s comprehension each set of experiments was performed on different groups of rats sacrificed at different time points. Since the number of experiments is considerable, a schematization or a table showing how exactly the total number of animals used in this study is subdivided among the various experiments is needed to better understand the study design.

Response: the total number use of animal is 168 and the experimental scheme were presented in “Methods” (Figure 10).

Figures:

It would be advisable to use a coherent color pattern throughout the whole manuscript. Here, some figures have colors while others are black and white. Please select a single style and use it for all the illustrations.

Response: Figures have been changed to Colorful pattern

Figure 4: please change “Contol” to “Control”

Response: We have corrected it

Title: The title is somehow complicated. A simpler title would be easier to comprehend for the readers.

Response: We have simplified the Title.

Round 2

Reviewer 1 Report

The authors have addressed all main points raised. Notably, the results in the modified Fig. 3, based on a larger sample size, seem more consistent.

The additional data confirming the knock-down of Pgc-1α are useful, and should be included as supplementary information in the publication.

While the changes to the figure, and particularly the inclusion of additional data, are very welcome, the graph style of the different figures is now very inconsistent. For the final submission, a consistent style (consistent colouring scheme and no hatching pattern) should be used throughout.

Some improvements of the English are already noticeable, and I trust that any remaining issues will be ironed out by MDPI during the editing process, as indicated by the authors.

Author Response

Response to Reviewer#1

1. The authors have addressed all main points raised. Notably, the results in the modified Fig. 3, based on a larger sample size, seem more consistent.

Response: Thank you for your comments.

2.The additional data confirming the knock-down of Pgc-1α are useful, and should be included as supplementary information in the publication.

Response: As your suggestions, we have added the data of knock-down of Pgc-1α in the supplementary information.

3. While the changes to the figure, and particularly the inclusion of additional data, are very welcome, the graph style of the different figures is now very inconsistent. For the final submission, a consistent style (consistent colouring scheme and no hatching pattern) should be used throughout.

Response: Thank you for your suggestion we have changed the graph style to keep consistent including colorful and style.

 4. Some improvements of the English are already noticeable, and I trust that any remaining issues will be ironed out by MDPI during the editing process, as indicated by the authors.

Response: Thank you. Our manuscript has been submitted to MDPI for English editing: English ID: english-22526.

Reviewer 2 Report

The authors have addressed the raised concerns and the manuscript presentation is improved. 

Author Response

Response to Reviewer 2#

  1. The authors have addressed the raised concerns and the manuscript presentation is improved

Response: We are very appreciated and thank you for your comments.

This manuscript is a resubmission of an earlier submission. The following is a list of the peer review reports and author responses from that submission.